# Plant–atmosphere and soil–atmosphere temperature differences and their impact on grain yield of super hybrid rice under different irrigation conditions

**Guiyuan Meng**[1,2☯]**, Rongqian Zheng**[1]**, Haiping Chen**[1]**, Guohui Ma**[2,3]**, Zhongwei Wei**[2,3]**, Guohong Xiang**[1]**, Jing Zhou**[1,2☯] *

1 College of Agriculture and Biotechnology, Hunan University of Humanities, Science and Technology, Loudi, Hunan Province, PR China, 2 State Key Laboratory of Hybrid Rice, Changsha, Hunan Province, PR China, 3 Hunan Hybrid Rice Research Center, Changsha, Hunan Province, PR China

☯ These authors contributed equally to this work.
* kiki010206@163.com

**Data Availability Statement:** All relevant data are within the paper and its Supporting Information files.

## Abstract

Continued drought during the late growth stage of super hybrid rice (SHR) markedly reduces yield, and management practices that use water more efficiently can contribute greatly to high and stable yields from SHR. The absolute temperature differences (ATDs) between the rice plant and the atmosphere and between the soil and the atmosphere are believed to be important determinants of grain yield. However, it has not previously been determined whether these ATDs have any effect on SHR yields under water-saving cultivation. A two-year field experiment involving two SHR varieties, Liangyoupeijiu (LYPJ) and Y-Liangyou 9000 (YLY900), evaluated the effects of reducing water supply from mid-booting to maturity on grain yield, canopy relative humidity (CRH), leaf area index (LAI), and ATDs between the ambient temperature and the leaf surface, panicles, canopy, and soil. Grain yield increased significantly under shallow water irrigation (SW), by 8.84% (YLY900) and 12.26% (LYPJ), but decreased significantly under mild water stress (MS, −20 to −30 kPa), by 14.36% (YLY900) and 9.47% (LYPJ), as well as severe water stress (SS, −40 to −50 kPa), by 35.06% (YLY900) and 28.74% (LYPJ). As water supply decreased, so did the CRH and the ATDs, with significant decreases under MS and SS. The temperature differences were significantly and positively correlated with grain yield ($P < 0.01$) in both cultivars. LAI was increased under SW conditions, but was significantly decreased under MS and SS. Our study suggests that the dual goal of saving water while maintaining high yield can be achieved by applying SW irrigation from mid-booting to maturity and by adopting cultivation measures that maintain high CRH and high plant–atmosphere and soil–atmosphere ATDs in order to alleviate water stress. YLY900 has a higher yield potential than LYPJ under SW conditions, suggesting that its wide cultivation may help achieve this dual goal.

**Funding:** This work was funded by funded by the National Natural Science Foundation of China (Grant no. 31701375), A National Key R&D Plan: Focus on Science and Technology Innovation for High Grain Efficiency (2018YFD0301002, 2018YFD0301004).

**Competing interests:** Conceptualization: Guiyuan Meng, Guohui Ma, Zhou Jing Funding acquisition: Guohui Ma, Zhongwei Wei Investigation: Guiyuan Meng, Rongqian Zheng, Haiping Chen, Zhongwei Wei, Guohong Xiang. Writing and editing: Haiping Chen, Guohong Xiang, Guiyuan Meng. No conflict of interest exits in the submission of this manuscript, and manuscript is approved by all authors for publication. The authors have declared that no competing interests exist.

## Introduction

Rice (*Oryza sativa* L.) is an important food crop around the world, especially in Asia, and the staple of about two-thirds of China's population [1, 2]. Many Asian countries are developing high-yielding rice varieties as a main research goal to ensure food security. Japan began research on super-high-yielding rice in 1980, and the International Rice Research Institute started breeding rice for super-high yields in 1989 [3]. China officially launched its super-rice breeding program in 1996, aiming to cultivate new rice varieties with high yield potential, good quality, and strong resistance [4, 5], and has made great progress over the past 20 years [6, 7]. By 2019, 132 super-high-yielding varieties of rice had been approved by the Ministry of Agriculture and were being grown over more than 70 million ha. Compared with non-super (check) varieties, super hybrid rice (SHR) varieties can increase grain yield by 15%–20% in field experiments and by about 10% in large-scale production [8–10]. Many agronomic and physiological traits contribute to the increased yield of these varieties, such as greater storage capacity, more grains per panicle [8–11], higher leaf area index and photosynthetic rate [10], and stronger lodging resistance, especially under high nitrogen levels [12, 13], compared with the best check varieties. However, these high yields have generally been achieved with copious irrigation, and it is not clear whether such high yields are possible under water deficits or water-saving irrigation.

Rice production is particularly water intensive, and the crop is therefore highly susceptible to deficits in soil water [14–17]. Approximately 50% of world rice production is affected by water scarcity to varying extents [18–20]. The effects of water deficits on rice include smaller grains, lower thousand-grain weight, lower seed-setting rate, and greater proportions of sterile spikelets, leading to markedly smaller yields [20–23]. Practicing judicious water management is considered an effective way to mitigate the adverse effects of water deficits on rice yield [24, 25], and careful monitoring of soil moisture is of great significance for proper management. One way to do this in a standing crop is by monitoring the temperature of various parts of the rice plant, as well as that of the soil, as these temperatures are affected by water deficits. Previous studies have shown that under drought, the canopy temperature increases, the absolute temperature difference (ATDs) between the canopy and the atmosphere falls, and the number of filled grains and grain yield decreases [26, 27]. When soil moisture decreases after the heading stage, leaf temperature increases with the drop in moisture, and yield decreases with a decrease in the difference between the temperatures of the atmosphere and leaf surface [28, 29]. As soil temperature is closely linked to water content, soil drought can cause a soil temperature rise [30, 31]. Canopy relative humidity (CRH) significantly decreases when watering is alternated with moderate or severe soil drying, and periods of severe soil drying significantly reduce grain yield [32]. These observations show that in rice, the CRH and the temperature of the plant organs and soil have a significant impact on plant growth and grain yield.

However, previous research on CRH and plant and soil temperatures has mainly been conducted with ordinary rice varieties, so it may be worthwhile to study the effects of water deficit on agronomy, yield, and plant and soil temperatures in SHR fields. At present, SHR researchers have paid inadequate attention to the effects of water deficits on CRH and on the plant–atmosphere and soil–atmosphere ATDs, although such information may prove crucial to obtaining higher yields from SHRs under reduced irrigation. To bridge this gap in information, the present study tested two SHR varieties in the field at four levels of irrigation. The specific objectives of this study were (1) to compare the effects of different levels of irrigation on grain yield, CRH, leaf area index (LAI), and plant–atmosphere and soil–atmosphere ATDs; (2) to determine the relationship between grain yield and changes in CRH and ATDs as affected by decreased water supply; and (3) to investigate the grain yield and yield component performance of SHR under different levels of irrigation.

## Materials and methods

### Field locations and treatments

The experiment was conducted in 2017 and 2018 at the experimental farm of the Hunan University of Humanities and Technology (27°12′ N, 112°31′ E, 170 m above sea level) in Loudi, Hunan Province, China. The top 20 cm of soil consisted of a clay loam with the following properties (average of samples taken each year): organic matter, 19.4%; available N, 93.54 mg kg$^{-1}$; available P, 21.68 mg kg$^{-1}$; available K, 123.7 mg kg$^{-1}$; pH, 6.8.

The tested SHR varieties were Liangyoupeijiu (LYPJ), a first-generation variety released in 1999, and Y-Liangyou 900 (YLY900), a fourth-generation variety released in 2015 [33], which are widely grown in southern China [34]; they were selected with the aim of comparing water deficit performance and yield between the newly elite and old SHR varieties. Pregerminated seeds were sown on a seedbed, and 30- or 31-day-old seedlings were transplanted to the experimental field (June 16, 2017, and June 20, 2018), with two seedlings to a hill and a hill spacing of 24 × 24 cm.

The field experiment was designed as a split-plot study, with each main plot being assigned to one of four irrigation treatments (three replicates per treatment) and divided into subplots for each of the two cultivars. Each plot was 18 m$^2$ and was separated from the adjacent plot by a cement ridge. The four treatments were as follows: the control or check (CK) treatment, a deep (8–10 cm) layer of water; the shallow water (SW) treatment, a shallow (1–3 cm) layer of water; the mild stress (MS) treatment, mild water stress (−20 to −30 kPa); and the severe stress (SS) treatment, severe water stress (−40 to −50 kPa). The treatments were applied from the fifth growth stage (booting) to maturity. Nitrogen was supplied as part of the basal dose of fertilizer (108 kg ha$^{-1}$), followed by another dose (54 kg ha$^{-1}$) seven days after transplanting and one more (108 kg ha$^{-1}$) at panicle emergence (the initiation of a differentiated apex). Phosphorus as superphosphate ($P_2O_5$, 135 kg ha$^{-1}$) was given as part of the basal dose, as was half (105 kg) of the total dose of potassium ($K_2O$, 210 kg ha$^{-1}$), with the remaining half (105 kg) given at panicle emergence.

Negative-pressure vacuum gauges (manufactured by the Nanjing Soil Research Institute of the Chinese Academy of Sciences) were installed to control the soil water potential. The gauges were read daily between 06:00 and 07:00, 12:00 and 13:00, and 17:00 and 18:00. If the reading was lower than the desired value, water was added manually in many small doses to maintain the soil water potential within the desired range.

The rice was grown in a natural environment before the treatments were applied; soil water was controlled only from booting to maturity. A movable plastic canopy with steel frames sheltered the crop from rain. Other management practices were the same as those followed for conventional management of high-yielding rice varieties.

### Experimental parameters and measurement methods

At the maturity stage, a 4-m$^2$ area of each plot was used to calculate grain yield, adjusted to a standard moisture content of $0.14 \times 10^{-3}$ $H_2O \cdot kg^{-1}$ fresh weight. Ten hills in that area were sampled diagonally to determine the yield components. Panicle number per square meter was determined by counting the panicles in the 10 hills. Panicles were threshed by hand, and filled spikelets were separated from unfilled ones using a blower. Three subsamples (30 g each) of filled spikelets were used to estimate the total number of filled spikelets, and the number of unfilled spikelets was determined by simple counting. The dry weight of spikelets (filled and unfilled) was determined after oven drying at 70°C to a constant weight. The percentage rate of seed setting was calculated by dividing the number of filled spikelets by the total number of spikelets and multiplying the result by 100.

During the period from the beginning of the treatments to maturity, three automatic temperature and humidity recorders (HOBO MX2302, Onset, Cape Cod, MA, USA) were installed in each plot. The height of the devices was set to match that of the panicle layer (two-thirds of plant height at that time), and changes in temperature and humidity in the canopy were recorded between 13:30 and 14:00. The temperature of the panicles and of the flag leaf was measured using an infrared thermometer (accurate to 0.1˚C; Raytek ST60+, Santa Cruz, USA), while soil temperature was measured using a soil thermometer at depths of 5 cm and 12 cm. Five plants were from each plot were selected for measurement, and the five measurements were averaged. Atmospheric temperature was measured with a wet-and-dry-bulb thermometer fixed vertically on a bracket at a height of 2 m from the ground. A leaf area meter (LI-3050C, Li-Cor, Lincoln, NE, USA) was used to determine the leaf area of the selected plants at filling and maturity. Observations were recorded during the grain-filling stage.

## Statistical analysis

Data were analyzed by performing three-way analysis of variance using Statistix ver. 8 (Analytical Software, Tallahassee, FL, USA). The means were compared using the least significant difference test.

## Results

### Climate conditions

The average daily maximum, minimum, and mean temperatures (Tmax, Tmin, Tmean) and the average relative humidity during the growing season (from transplanting to maturity) were 30.6˚C, 22.9˚C, 26.7˚C, and 77.8% in 2017 and 30.5˚C, 22.1˚C, 26.3˚C, and 76.3% in 2018 (Fig 1). Large differences in daily temperatures were observed in each year during the grain-filling stage, and daily Tmax, Tmin, and Tmean were higher in 2017 (by 1.1˚C, 0.8˚C, and 0.93˚C, respectively) than in 2018.

### Grain yield and yield components

There were significant differences in yield between varieties and between treatments (Tables 1 and 2). Compared with control (CK), the SW treatment produced a yield that was greater by 101.51–118.91 g m$^{-2}$ in LYPJ and by 95.21–107.17 g m$^{-2}$ in YLY900. The average yield decreased significantly with water stress: compared with CK, LYPJ yield decreased by 9.48% under MS and by 28.75% under SS, while the corresponding decreases for YLY900 were 14.36% and 35.06% (values are averages of the two years). Grain yields in 2017 were higher than in 2018 by 4.58% in LYPJ and 3.87% in YLY900. Across all treatments, YLY900 produced higher grain yields than LYPJ (by 188.08 g m$^{-2}$ in 2017 and 186.58 g m$^{-2}$ in 2018).

Spikelets per panicle, seed setting rate, and grain weight also varied significantly among treatments, and the differences between the two varieties were mainly due to differences in these yield components (Table 2). The number of spikelets per panicle was significantly greater for SW than for CK, by 7.08% in LYPJ and 4.70% in YLY900. Water stress, whether mild or severe, reduced the number of spikelets per panicle and grain weight significantly, whereas seed setting rate decreased significantly only in plants under severe water stress. Number of panicles and grain weight were significantly greater in LYPJ than in YLY900: the number of panicles for LYPJ was higher by 16.34% in 2017 and 10.88% in 2018, and the corresponding numbers for grain weight were 3.84% and 3.67%, respectively. The number of spikelets per panicle in LYPJ was significantly lower than in YLY900, on average by 32.81% in 2017 and 29.89% in 2018.

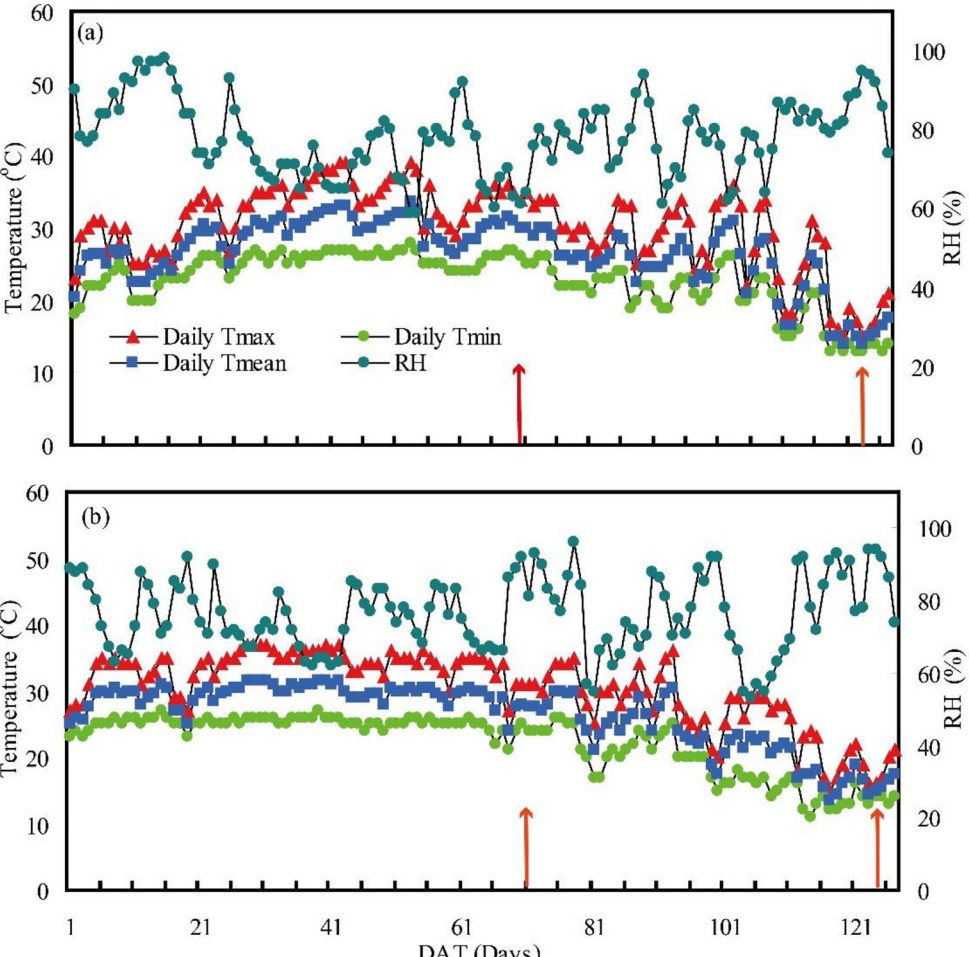

**Fig 1.** Maximum temperature (Tmax), minimum temperature (Tmin), average temperature (Tmean), and relative humidity (RH) during the growing season in 2017 (a) and 2018 (b) in Loudi, Hunan Province, China. The two red arrows indicate the start and end of the grain-filling stage (70–123 days after transplanting in 2017 and 71–125 days after transplanting in 2018).

**Table 1. Analysis of variance (ANOVA) of F-values for grain yield (GY), panicle number (P), spikelets per panicle (SP), seed-setting rate (SR), and grain weight (GW).**

| ANOVA | GY | P | SP | SR | GW |
|---|---|---|---|---|---|
| Year (Y) | ns | 100.98** | ns | ns | 17.93* |
| Variety (V) | 258.67** | 769.29** | 537.23** | ns | 475.42** |
| Soil water (S) | 245.01** | ns | 146.72** | 303.89** | 215.41** |
| Y× V | ns | 49.78** | 15.62* | ns | ns |
| Y×S | 11.05* | ns | ns | ns | ns |
| V×S | 80.06** | 4.17* | 51.92** | ns | ns |
| V× S×Y | ns | ns | ns | ns | ns |

ns, not significant; * and ** denote significance at the 0.05 and 0.01 probability levels, respectively.

**Table 2. Effect of water treatments on grain yield and its components in 2017 and 2018.**

| Year | Variety | Treatment | P (m²) | SP (panicle⁻¹) | SR (%) | GW (mg) | GY (g m⁻²) |
|------|---------|-----------|--------|----------------|--------|---------|------------|
| **2017** | LYPJ | CK | 248.96 a | 167.74 b | 83.03 ab | 26.56 a | 920.94 b |
| | | SW | 250.90 a | 178.46 a | 86.72 a | 26.78 a | 1039.85 a |
| | | MS | 252.20 a | 156.59 c | 80.37 b | 26.15 b | 829.99 c |
| | | SS | 249.60 a | 138.68 d | 73.49 c | 25.36 c | 645.11 d |
| | | Mean | **250.42 A** | **160.37 B** | **80.90 A** | **26.21 A** | **851.62 B** |
| | YLY900 | CK | 217.92 a | 256.86 b | 82.69 ab | 25.56 ab | 1183.06 b |
| | | SW | 215.98 a | 269.43a | 84.65 a | 25.95 a | 1278.27 a |
| | | MS | 213.38 a | 234.41c | 79.23 b | 25.16 b | 997.08 c |
| | | SS | 213.68 a | 194.06 d | 74.21 c | 24.27 c | 746.85 d |
| | | Mean | **215.24B** | **238.69 A** | **80.20 A** | **25.24 B** | **1039.70 A** |
| **2018** | LYPJ | CK | 235.18 a | 170.35 b | 82.53 ab | 26.45 a | 874.54 b |
| | | SW | 234.53 a | 183.58 a | 85.45 a | 26.53 a | 976.05 a |
| | | MS | 237.13 a | 162.06 c | 79.93 b | 25.89 b | 795.25 c |
| | | SS | 236.48 a | 143.48 d | 74.29 c | 25.14 c | 633.70 d |
| | | Mean | **235.83 A** | **164.87 B** | **80.55 A** | **26.00 A** | **814.36 B** |
| | YLY900 | CK | 212.20 a | 251.50 b | 81.75 ab | 25.49 a | 1112.09 b |
| | | SW | 214.80 a | 262.85 a | 84.29 a | 25.62 a | 1219.26 a |
| | | MS | 212.85 a | 229.85 c | 79.61 b | 24.84 b | 967.47 c |
| | | SS | 210.90 a | 196.48 d | 73.58 c | 24.35 c | 742.43 d |
| | | Mean | **212.69 B** | **235.17A** | **79.81 A** | **25.08 B** | **1000.94 A** |

Values followed by different letters within the same column for a cultivar are significantly different at *P* = 0.05. CK, deep water layer; SW, shallow water layer; MS, mild water stress; SS, severe water stress.

## Temperature differences between plants and atmosphere

Leaf surface, panicle, and canopy temperatures were consistently lower than atmospheric temperatures. ATD was highest between the canopy and the atmosphere (C-A) and lowest between the panicles and the atmosphere (P-A), with the ATD between the leaf surface and the atmosphere (L-A) in between. All aforementioned ATDs decreased with reduced moisture levels, and the reduction was significant under MS and SS treatments in both varieties, with average decreases as follows: MS, 0.71˚C (C-A), 0.65˚C (L-A), 1.16˚C (P-A); SS, 1.05˚C (C-A), 1.16˚C (L-A), 1.68˚C (P-A) (Fig 2).

## Temperature differences between soil and atmosphere

The ATD between the soil and the atmosphere was lower for YLY900 than for LYPJ (Fig 3). The values decreased significantly under water stress. In LYPJ, the temperature at a depth of 5 cm decreased by 0.56˚C under MS and by 0.35˚C under SS. The corresponding values for YLY900 were 0.98˚C and 0.73˚C, respectively. At a depth of 12 cm, there were significant decreases in soil–atmosphere ATD for SW, MS, and SS, and these decreases were greater than those at 5 cm, with the average increase (average of both years) being 0.59˚C in YLPJ and 0.56˚C in YLY900.

## Canopy relative humidity

Decreasing water supply reduced CRH significantly: for LYPJ, the decrease was 11.48% under MS and 20.43% under SS, while for YLY900, the corresponding figures were 12.34% and 20.88% (Fig 4). For any given treatment, CRH was lower for YLY900 than LYPJ. In addition, CRH was higher in 2018 than in 2017, by 5.96% for LYPJ and 6.08% for YLY900.

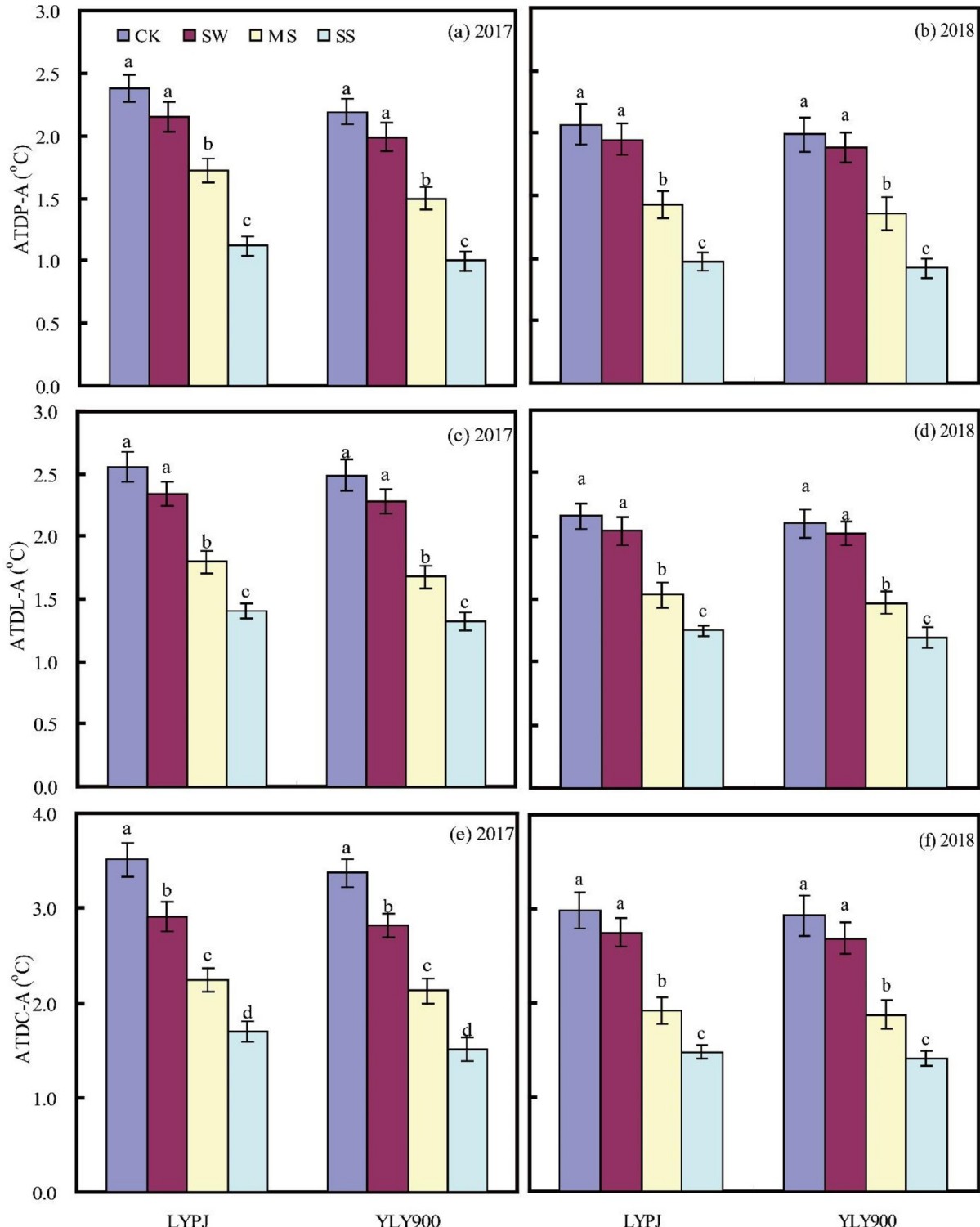

**Fig 2.** Absolute temperature differences between panicle and atmosphere (ATDP–A) (a and b), between leaf surface and atmosphere (ATDL–A) (c and d), and between canopy and atmosphere (ATDC–A) (e and f) in 2017 and 2018. CK, check; SW, shallow water; MS, mild stress; SS, severe stress; LYPJ, Liangyoupeijiu; YLY900, Y-Liangyou 900.

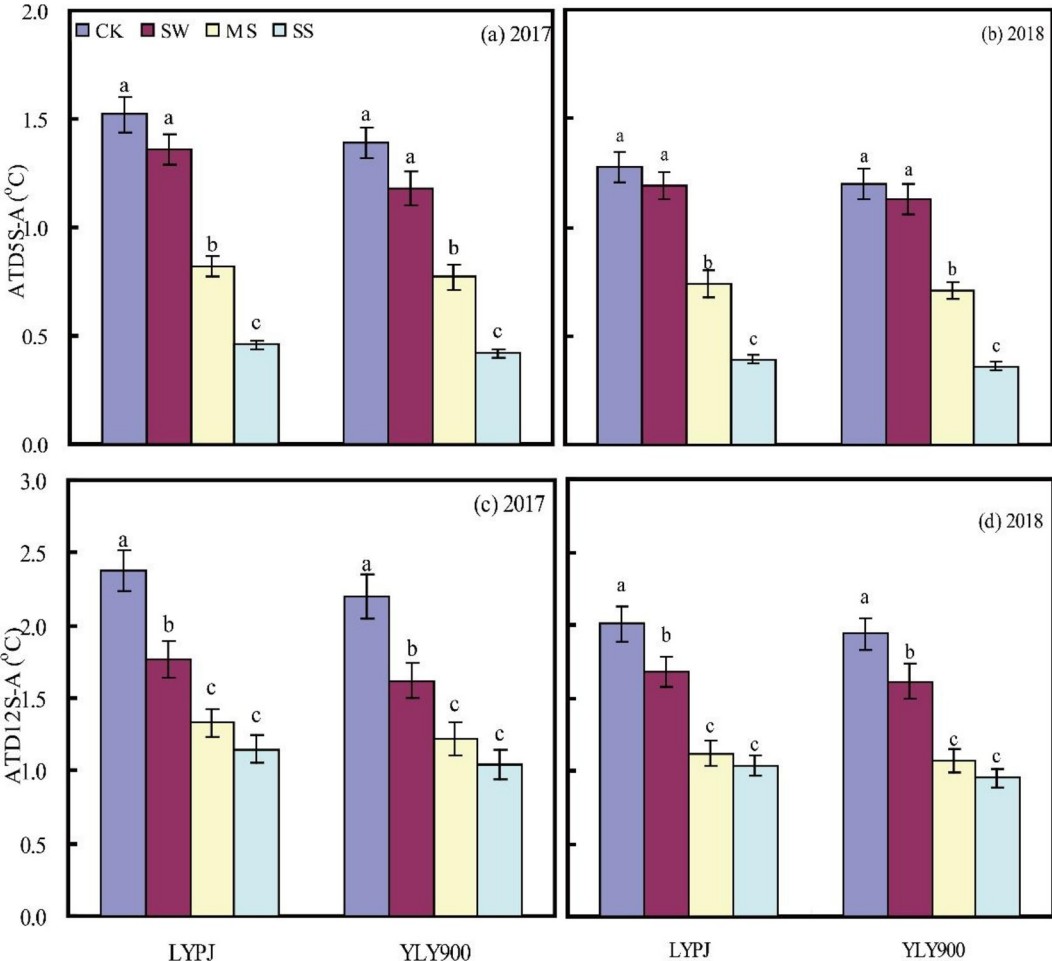

**Fig 3.** Absolute temperature differences between soil at depths of 5 cm and atmosphere (ATD5S-A) (a and b), and between 12 cm and atmosphere (ATD12S-A) (c and d) in 2017 and 2018. CK, check; SW, shallow water; MS, mild stress; SS, severe stress; LYPJ, Liangyoupeijiu; YLY900, Y-Liangyou 900.

### Leaf area index (LAI)

Leaf area index (LAI) at the filling and maturity stages was affected by water stress; the values did not significantly increase under SW conditions, but decreased significantly under MS and SS conditions (Fig 5). In LYPJ, LAI at grain filling was decreased by 0.20 under MS and 0.45 under SS, while at maturity, it was decreased by 0.28 under MS and 0.51 under SS. In YLY900, the corresponding figures at grain filling were 0.24 (MS) and 0.43 (SS), and those at maturity were 0.21 (MS) and 0.45 (SS).

### Correlation analysis

Grain yield was significantly and linearly correlated in both years with nearly all parameters of interest, namely the ATDs between plant parts and the atmosphere (L-A, P-A, and C-A) (Fig 6), the ATDs between the soil and the atmosphere (S-A) at depths of 5 cm and 12 cm, and the CRH (Fig 7). However, the degree of correlation varied with the cultivar, being stronger in YLY900 than in LYPJ. Correlation coefficient ($R^2$) values were as follows for YLY900 and LYPJ, respectively: L-A, 0.83 and 0.74; P-A, 0.87 and 0.79; C-A, 0.80 and 0.68; S-A at 5 cm, 0.88 and 0.80; S-A at 12 cm, 0.60 and 0.45; CRH, 0.72 and 0.62 (Figs 6 and 7).

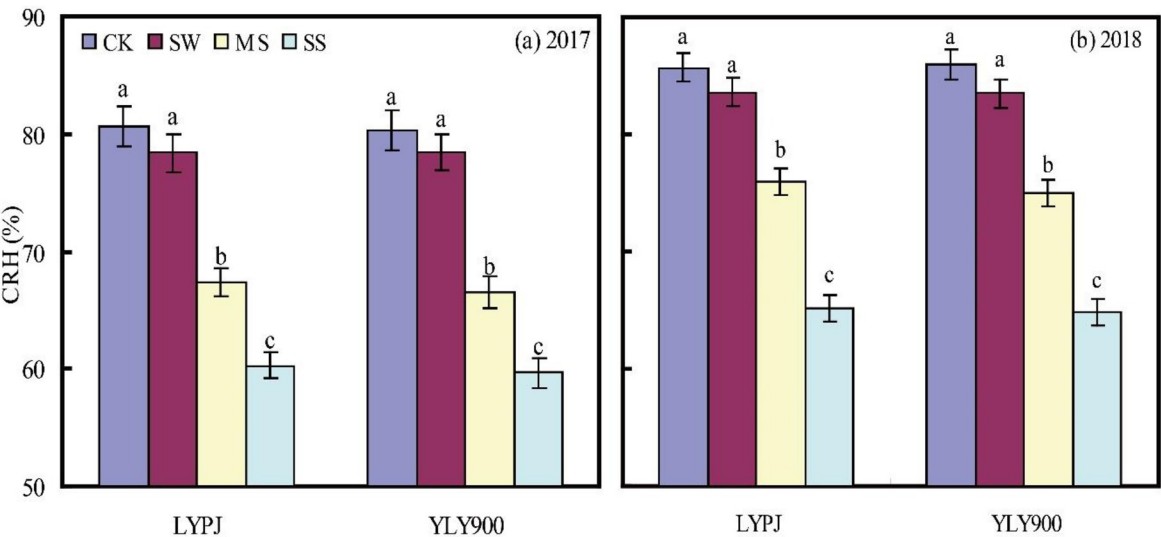

**Fig 4.** Canopy relative humidity (CRH) under different treatments in 2017 (a) and 2018 (b). CK, check; SW, shallow water; MS, mild stress; SS, severe stress; LYPJ, Liangyoupeijiu; YLY900, Y-Liangyou 900.

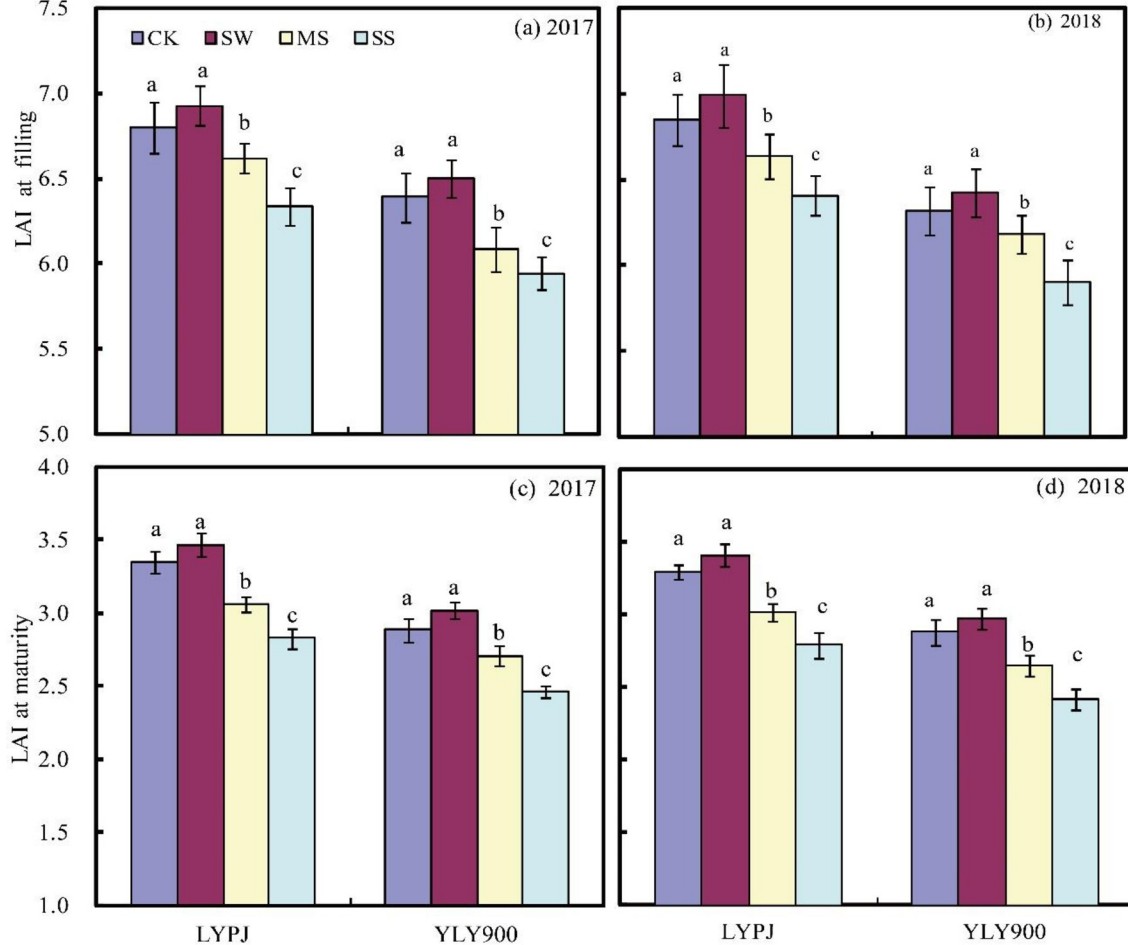

**Fig 5.** Leaf area index (LAI) at filling (a and b) and maturity (c and d) in 2017 and 2018. LYPJ, Liangyoupeijiu; YLY900, Y-Liangyou 900.

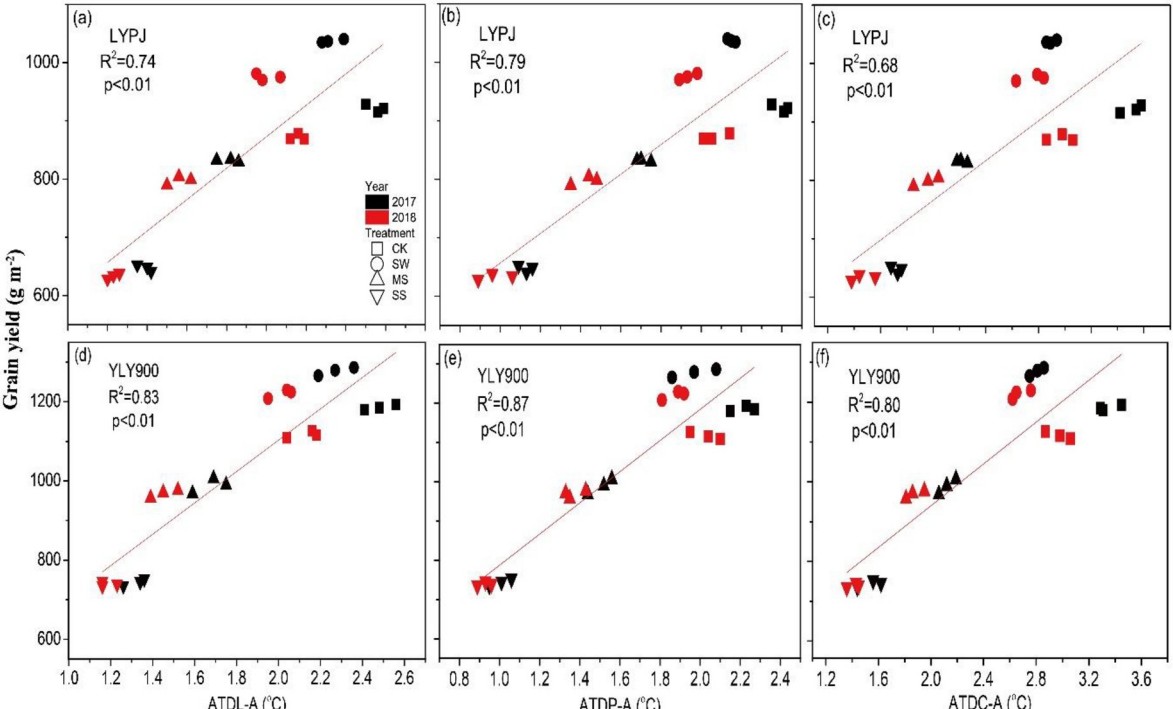

**Fig 6.** Correlations of grain yield with absolute temperature differences between the leaf surface and atmosphere(ATD L–A), between the panicle and atmosphere (ATD P–A), and between the canopy and atmosphere (ATD C–A) in Liangyoupeijiu (LYPJ) (a, b, c) and Y-Liangyou 900 (YLY900) (d, e, f).

## Discussion

The superiority of SHR over check varieties in terms of biomass production and grain yield is well documented [11, 35, 36]. Over the past few decades, SHR has been giving record yields under optimal cultivation and management conditions in China. For example, two SHR cultivars, YLY900 and Chaoyou 1000, have given record yields of 15.4 t ha$^{-1}$ and 14.1 t ha$^{-1}$, respectively [37–39]. However, such high yields can only be obtained in a few specific areas with conducive conditions in terms of sunlight and temperature and with ample fertilizer and water. In other areas, grain yields have been relatively low because of drought, inadequate irrigation, low temperatures, and cloudy days during the growing season. Previously, yield and other factors related to SHR performance have rarely been compared under different levels of soil moisture.

The most critical stages for hybrid rice yield are tillering, booting, and flowering. Water stress at any of these stages can have marked adverse effects on the major yield components [40]. In the present study, the grain yield of SHR was significantly higher under SW irrigation than under deep-water irrigation (Table 2), suggesting that SHR might have higher yield potential if water-saving irrigation is applied from mid-booting to maturity. There are several possible explanations for this observation. Under the SW treatment, SHR produced more spikelets per panicle, had a better seed-setting rate, and had a greater grain weight than under the other treatments (Table 2). These results suggest that the aforementioned yield components are the key factors that determine grain yield when the plant is subjected to water stress from mid-booting to maturity. Thus, an SHR variety capable of producing reasonably high values for these yield components under water stress would give a greater grain yield [20, 41]. SHR under SW also showed a higher LAI at the late growth stage (Fig 5), indicating that the

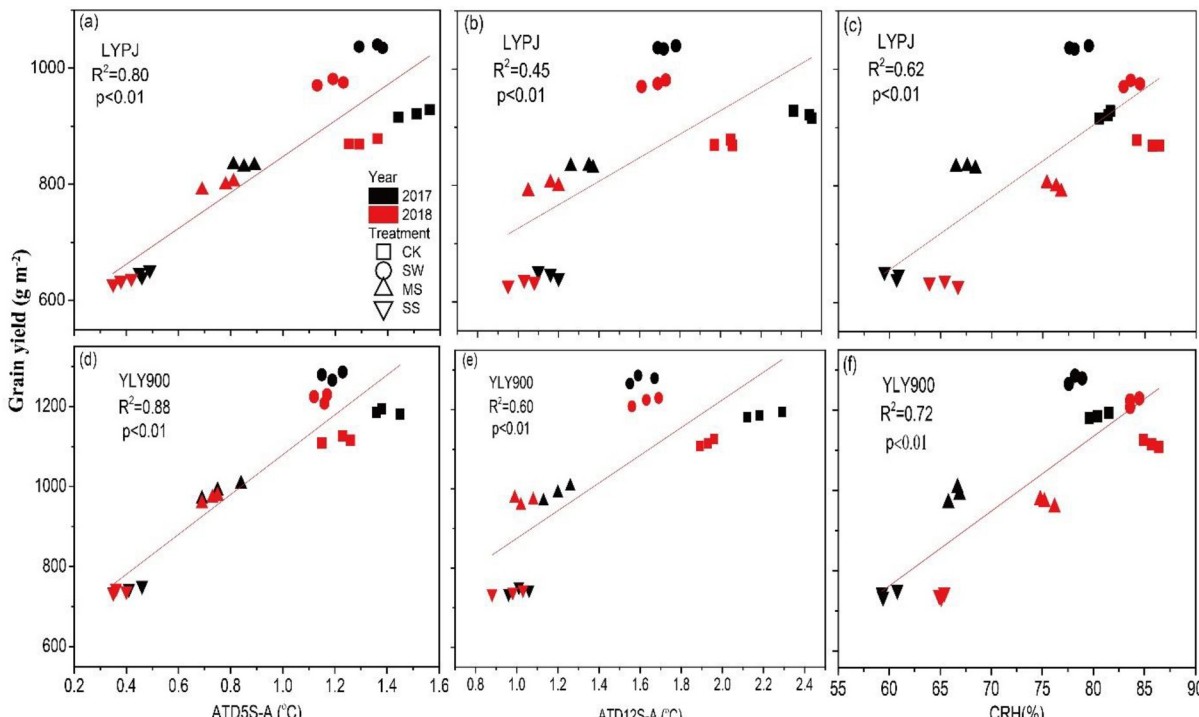

**Fig 7.** Correlations of grain yield with absolute temperature differences between the soil at depths of 5 cm and atmosphere(ATD 5S–A), and between 12 cm and atmosphere (ATD 12S–A), in Liangyoupeijiu (LYPJ) (a, b, c) and Y-Liangyou 900 (YLY900) (d, e, f).

plants had a stronger photosynthetic capacity and accumulated more photosynthetic products than in other treatments, resulting in a higher number of spikelets per panicle, better seed setting rate, and greater grain weight. It is believed that increasing SHR yields may require a large LAI to improve canopy quality and increase radiation use efficiency [10, 20]. In our study, the new elite variety, YLY900, had better yield potential under the same cultivation and irrigation conditions than the old variety, LYPJ, with a significantly higher average yield for all four treatments (Table 2). The main reason for the higher yield was a higher number of spikelets per panicle, with our results indicating that this variety has strong adaptability to water deficits and should be promoted and cultivated more broadly.

Previous studies that examined ambient temperature focused more on its relationship to spikelet development, transpiration and plant canopy temperature, paying little attention to its relationship with the temperature of different plant parts and the correlations of these differences with grain yield [42–45]. Our research showed that leaf, panicle, and canopy temperatures tended to be lower than the atmospheric temperature. Among these, the canopy had the lowest temperature, leaf surface temperature was intermediate, and panicle temperature was closest to the ambient temperature (Fig 2), which can be explained by the greater exposure to radiation experienced by the upper parts of a plant compared with parts closer to the ground [46]. The study also found that with decreased water supply, the ATDs between plant parts and the atmosphere decreased. This may be because low soil moisture affects the transpiration rate, leading to an increase in plant temperature [45]. The results also showed that the ATDs were associated with different effects on SHR yield. The moderate decrease in ATDs under SW irrigation was associated with good grain development, while the significant decrease under MS and SS had a clear influence on the transpiration rate, which in turn affected the number of grains per panicle, seed setting rate, and grain weight, with these yield components

also being factors in the observed high yield under SW and low yield under MS and SS. Previous studies have also reported that such changes affect grain development [26, 27, 47].

Soil temperature is particularly important to the rate and direction of energy and mass exchange (including evaporation and aeration) between soil and the atmosphere, and these phenomena, particularly soil moisture evaporation and heat transfer, affect all aspects of crop production. In our study, decreasing levels of soil moisture were associated with decreased ATD between soil and the atmosphere, indicating that as soil moisture content decreased, soil temperature increased (Fig 3), an observation consistent with two earlier studies [30, 31]. ATD was lower at a depth of 5 cm than at 12 cm under all treatments, which can be attributed to the upper soil layer's faster increase in temperature upon exposure to solar radiation and the greater effect exerted on it by the ambient temperature. Humidity is another important factor affecting fertilization and grain formation in rice [32, 48]. In the present study, CRH decreased as the soil moisture content decreased, by approximately 10% under MS and by approximately 20% under SS (Fig 4). The decrease in soil moisture likely affected plant transpiration, leading to the change in CRH, which in turn affected the development of the spikelets and grain [45]. The ultimate result was significantly fewer spikelets per panicle, poor seed set, and lighter grains.

We found a significant ($P < 0.01$) linear correlation between grain yield and each of the study's parameters of interest when the plants were under water stress (Figs 6 and 7). These results suggest that decreased ATDs between rice plants and the atmosphere and between soil and the atmosphere can affect the accumulation of assimilates and decrease grain yield [49, 50]. Thus, lower ATD and lower CRH appear to be partly responsible for the decrease in grain yield under water stress, and maintaining higher ATD and CRH from booting to maturity can reverse the decrease in grain yield. Judicious irrigation can lower the temperature of the rice canopy, leaf surface, and panicle, thus increasing the temperature difference between the plant and the atmosphere and increasing rice yield. This conclusion was obtained in fields in southern Hunan Province, China; considering that crop yield may be affected by variety, climate, cultivation measures, and other factors, further research is necessary to determine whether the result can be generalized to a wider range of conditions, rice varieties, and crops.

## Conclusion

As the level of water decreased from mid-booting to maturity stage, CRH decreased, as did ATDs between the atmosphere and the leaf surface, panicles, crop canopy, and soil. The decrease in these parameters was positively correlated with grain yield ($P < 0.01$). This study provides a scientific assessment of temperature differences between rice plant parts and the atmosphere, as well as their impact on yield, and contributes key information for soil moisture monitoring and water conservation during the grain-filling stage. Compared with deep-water irrigation, SHR has higher yield potential under SW conditions, which is mainly attributable to higher LAI, greater photosynthetic capacity, and improved yield components (spikelets per panicle, seed setting rate, and grain weight). Our results show that adoption of SW-based irrigation from mid-booting to maturity holds great promise for achieving the dual goal of increasing yield and saving water in SHR cultivation. The new elite variety YLY900 has a higher yield performance than the old variety LYPJ under the same cultivation and irrigation conditions, making it suitable for wider adoption. These data are also valuable for improving planting efficiency and farmers' income.

## Supporting information

**S1 Data.**
(XLS)

## Acknowledgments

We thank Dr. Ren-Yan Duan, Ying Song and anonymous reviewers for comments on the article.

## Author Contributions

**Conceptualization:** Guiyuan Meng, Guohui Ma, Jing Zhou.

**Funding acquisition:** Guohui Ma, Zhongwei Wei.

**Investigation:** Guiyuan Meng, Rongqian Zheng, Haiping Chen, Zhongwei Wei, Guohong Xiang.

**Writing – original draft:** Haiping Chen.

**Writing – review & editing:** Guiyuan Meng, Guohong Xiang, Jing Zhou.

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
