## [Decision Letter · Decision Letter 0]

23 Sep 2020

PONE-D-20-26758

High Temperature Difference bewteen Rice Organs ,Soil and Atomsphere Improves Yield of Super Hybrid Rice during Grain-Filling Stage

PLOS ONE

Dear Dr. Zhou,

Thank you for submitting your manuscript to PLOS ONE. After careful consideration, we feel that it has merit but does not fully meet PLOS ONE’s publication criteria as it currently stands. Therefore, we invite you to submit a revised version of the manuscript that addresses the points raised during the review process.

We look forward to receiving your revised manuscript.

Kind regards,

Vincent Vadez

Academic Editor

PLOS ONE

Journal Requirements:

'Conceptualization: Guiyuan Meng, Guohui Ma, Zhou Jing

Funding acquisition: Guohui Ma, Zhongwei Wei

Investigation: Guiyuan Meng, Rongqian Zheng, Haiping Chen, Zhongwei Wei, Guohong Xiang.

Writing and editing: Haiping Chen, Guohong Xiang, Guiyuan Meng.

No conflict of interest exits in the submission of this manuscript, and manuscript is approved by all authors for publication'

a. Please complete your Competing Interests statement to state any Competing Interests.

If you have no competing interests, please state "The authors have declared that no competing interests exist.", as detailed online in our guide for authors at http://journals.plos.org/plosone/s/submit-now

Reviewers' comments:

Reviewer's Responses to Questions

**Comments to the Author**

1. Is the manuscript technically sound, and do the data support the conclusions?

Reviewer #1: Partly

Reviewer #2: No

2. Has the statistical analysis been performed appropriately and rigorously? 

Reviewer #1: Yes

Reviewer #2: No

3. Have the authors made all data underlying the findings in their manuscript fully available?

Reviewer #1: Yes

Reviewer #2: No

4. Is the manuscript presented in an intelligible fashion and written in standard English?

Reviewer #1: No

Reviewer #2: Yes

5. Review Comments to the Author

Reviewer #1: This is an interesting study where lots of detailed data on specific traits like plant organ temperature were measured which linked to yield parameters in rice. However, the manuscript can’t be accepted in a present form to be published in an international journal. The whole manuscript needs a rigorous revision for several grammatical and typographical errors.

On scientific aspect,

(i) the study had neither raised a strong scientific hypothesis in introduction nor showed what is novel about the results in discussion.

(ii) Introduction needs significant revision as it’s not to the point.

(iii) Major part of discussion just repeated results instead of explaining the novelty of each result.

(iv) Two super hybrid rice varieties were assessed and the reason behind the selection was their difference in release time. Apart from that there was no note on how they behaved for stress previously or what authors expect while assessing an old and a new variety.

(v) Whether the results from this study works just for China or to a broader rice/ crop community, if so make a mention on (something like tropical, temperate, Mediterranean regions)

Reviewer #2: Connecting non-limiting irrigation to lower soil and canopy temperature and greater yield is not particularly interesting, which isn’t necessarily a problem, but the lack of precision in connecting causality among irrigation, canopy temperatures and yield is problematic. I find this manuscript to be lacking basic physiologic rigor.

Presenting figs 5 and 6 as a regression is a bit misleading because there are 4 discreet treatment resulting in different temperature differentials, not a continuous distribution. Barplots could have been used for Fig 5 and 6, as in the other figs. That the 1-3cm irrigation treatment sometimes performed better than the 8-10cm irrigation treatment suggests less water can be used. But rather than highlight this, the discussion focuses on how irrigation is associated with reduced canopy temperature and higher yield.

Furthermore, there seems to be some confusion in the assertion that greater temperature difference between “plant organs” and the atmosphere resulted in higher yield, rather than being an indication of greater transpiration that likely permits greater C fixation. Another incongruous way this is stated is that temperature affects the “rates of transpiration and grain filling.” The literature cited (47 and 48) regarding the relationship between lower canopy temperature and performance are not the most appropriate. One of the references cited for the link between high temperature and reduced fertility corresponds to how elevated atmospheric temperatures can reduce pollen fertility. The other reference is a general review of photosynthesis and temperature from 1980. The role of transpiration in affecting temperature of leaf and panicle seems to have been ignored.

Another problematic link is in asserting that “lower canopy relative humidity led to greater transpiration, which depleted water content in leaves and stems.” Firstly, I see no indication that transpiration was actually directly measured, neither any indication that leaf and stem water content were measured. More problematically, there seems to be some confusion in that water needed for “proper development of the panicle and seed” comes from the leaves and stems rather than the soil.

In the conclusion, the comparison presented has to do with the difference between the 1-3cm of standing water treatment and the water stress treatments, while the “check” treatment with 8-10cm of standing water is ignored.

How was comparing monitoring time possible if temperature and humidity were only recorded from 13:30-14:00, as stated in the methods?

How is the “best monitoring time” determined? Specify why it was best, greatest differences among treatments, lowest CV within a treatment…

Specific issues:

Introduction: specify to what the rice has a “strong resistance” e.g. biotic, abiotic…

How are vacuum negative pressure gauges different from tensiometers? At what depth were they installed?

Leaf area per plot, soil coverage over time should have been reported to separate the effect of soil coverage from irrigation.

“Mild stress” not “mind stress”

Supporting data is not visible.

6. PLOS authors have the option to publish the peer review history of their article (what does this mean?). If published, this will include your full peer review and any attached files.

Reviewer #1: No

Reviewer #2: No

---

## [Author Response · Author response to Decision Letter 0]

17 Nov 2020

Response to Reviewers

Reviewer 1: 

(i) the study had neither raised a strong scientific hypothesis in introduction nor showed what is novel about the results in discussion.

The results were rearranged with emphasis on novelty.

(ii) Introduction needs significant revision as it’s not to the point.

The introduction has been greatly revised, trying to get to the point.

(iii) Major part of discussion just repeated results instead of explaining the novelty of each result.

The novelty of the results is explained in the discussion part.

(iv) Two super hybrid rice varieties were assessed and the reason behind the selection was their difference in release time. Apart from that there was no note on how they behaved for stress previously or what authors expect while assessing an old and a new variety.

The two varieties were selected to evaluate the performance of water deficit and yield of elite(YLY900) and old varieties(LYPJ),and promote better application(Added in text).

(v) Whether the results from this study works just for China or to a broader rice/ crop community, if so make a mention on (something like tropical, temperate, Mediterranean regions)

This conclusion was obtained under the test in southern Hunan provinces China. Considering that rice yield is affected by variety, climate, posters, cultivation measures and other factors, whether the result is suitable for a wider rice / crop community needs to be further explored.

Reviewer 2:

1、Presenting figs 5 and 6 as a regression is a bit misleading because there are 4 discreet treatment resulting in different temperature differentials, not a continuous distribution. Barplots could have been used for Fig 5 and 6, as in the other figs.

The opinions of the expert are very reasonable. According to the need of strict control of plant physiology, the temperature difference and yield regression should be continuously distributed, which requires the continuity of water gradient treatment in the experiment. Under natural ecological conditions, the temperature difference and yield distribution should have good continuity. However, according to the water demand, cultivation practice and preliminary preparation test of super rice after flowering, if the continuous water control treatment is set in the field test, the temperature difference under the similar treatment is less, but the operation intensity is increased. Field experiments are affected by many factors, such as environment, climate, cultivation and so on. Therefore, considering the effectiveness and operability, only four key water treatments, namely shallow water layer(1–3 cm), deep water layer(8–10 cm), mild water stress (−20 to −30 kPa), and severe water stress (−40 to −50 kPa), were set up to explore the change of temperature difference and its impact on yield under water control treatment. Although the correlation between temperature difference and yield was not well continuously distributed under the four key water treatments, it reflected a close correlation to a certain extent. I do this graph just to express the correlation. The original intention of the mapping is to refer to the article published by Liu on the journal “field crop research”(Relationships between grain yield and intercepted photosynthetically active radiation (IPAR) and radiation use efficiency (RUE) of three super rice varieties under different nitrogen levels).( Liu K, Yang R, Deng J, Huamg LY, Wei ZW, Ma GH, Tian XH. High radiation use efficiency improves yield in the recently developed elite hybrid rice Y-liangyou 900. Field Crops Res. 2020; 253:107804. http://doi:10.1016/j.fcr.2020.107804)

 If figures 5 and 6 are made into column chart, the abscissa is the temperature difference under four different water treatments, and the ordinate is the yield, but it is actually the yield difference diagram among different water treatments (the columnar shape is the yield difference, even if the abscissa shows the temperature difference data), so it is repeated with Fig.2, Fig.3 and Fig.4.

2、 That the 1-3cm irrigation treatment sometimes performed better than the 8-10cm irrigation treatment suggests less water can be used. But rather than highlight this, the discussion focuses on how irrigation is associated with reduced canopy temperature and higher yield.

Super rice has the advantage of water saving and higher yield ubder 1-3cm irrigation than that of 8-10cm irrigationt, and which is emphasized and explained in the second paragraph of the discussion.

3、Furthermore, there seems to be some confusion in the assertion that greater temperature difference between “plant organs” and the atmosphere resulted in higher yield, rather than being an indication of greater transpiration that likely permits greater C fixation.

The expression that greater temperature difference between “plant organs” and the atmosphere resulted in higher yield is not appropriate. So,the subject and related contents has made appropriate revision, and expounds the possible change of transpiration caused by temperature difference, and then the impact on yield.

4、Another incongruous way this is stated is that temperature affects the “rates of transpiration and grain filling.” The literature cited (47 and 48) regarding the relationship between lower canopy temperature and performance are not the most appropriate.

The expert opinion is very reasonable, and the inappropriate part has been deleted.

5、One of the references cited for the link between high temperature and reduced fertility corresponds to how elevated atmospheric temperatures can reduce pollen fertility. The other reference is a general review of photosynthesis and temperature from 1980. The role of transpiration in affecting temperature of leaf and panicle seems to have been ignored.

The effects of different water treatments on transpiration and organ temperature have been explained.

6、Another problematic link is in asserting that “lower canopy relative humidity led to greater transpiration, which depleted water content in leaves and stems.” Firstly, I see no indication that transpiration was actually directly measured, neither any indication that leaf and stem water content were measured. More problematically, there seems to be some confusion in that water needed for “proper development of the panicle and seed” comes from the leaves and stems rather than the soil.

After careful consideration of the experts' opinions, the revision was made.

7、How was comparing monitoring time possible if temperature and humidity were only recorded from 13:30-14:00, as stated in the methods?

How is the “best monitoring time” determined? Specify why it was best, greatest differences among treatments, lowest CV within a treatment.

The humidity canopy and temperature difference between leaf, panicle and atmosphere of super rice at 12:0, 12:30, 13:00, 13:30, 14:00, 14:30 and 15:00 were compared in preliminary test. It was found that 13:30-14:00 was the maximum temperature difference of each water treatment of super rice. Therefore, it can be considered that the temperature difference of this period can better reflect the soil moisture status.

The canopy humidity and temperature difference between plant organs, soil and atmosphere decreased with the reduction of soil water. Under mild (MS)and severe water stress(SS), the indexes were significantly less than those of deep water irrigation(CK) . Except that the temperature difference between 12 cm soil and atmosphere was not obvious, the other indexes were significantly different between MS and SS condion.

The coefficient of variation was not obvious in the same treatment. The minimum CV of temperature difference between plant organs and atmosphere was 0.02.

8、Specific issues:

(1) Introduction: specify to what the rice has a “strong resistance” e.g. biotic, abiotic…

Super rice is a rice variety bred by combining ideal plant type and heterosis utilization. It has the characteristics of moderate tillering, straight sword leaf, hard stem, lodging resistance and wide adaptability,and the physiological mechanism of high photosynthetic efficiency, strong root activity, source sink flow coordination, and has the genetic basis of stress resistance and disease resistance. (relevant content has been added in the introduction)

(2)How are vacuum negative pressure gauges different from tensiometers? At what depth were they installed?

Vacuum negative pressure gauge is based on atmospheric pressure, which is used to measure less than atmospheric pressure; it is applicable to measure the vacuum pressure of liquid and gas without explosion, crystallization, solidification, and corrosion of copper and copper alloy.

Tensiometer is the use of negative pressure meter to measure soil moisture, is a practical means to study soil water movement from the perspective of energy. When there is a pressure difference between the measured soil water potential and the tensiometer, the water in the tensiometer penetrates into the measured soil until it reaches the equilibrium state. At this time, the value displayed by the pressure gauge is the soil water potential of the tested soil. In this experiment, the ceramic probe of tensiometer was fixed at soil depths of 5 cm and 12 cm respectively.

(3)Leaf area per plot, soil coverage over time should have been reported to separate the effect of soil coverage from irrigation.

The leaf area index (LAI) at grain filling and mature stage was plotted, and the effect of different irrigation on LAI was analyzed.

(4)“Mild stress” not “mind stress”

The error has been revised.

---

## [Editor Report · Decision Letter 1]

20 Nov 2020

PONE-D-20-26758R1

Plant–atmosphere and soil–atmosphere temperature differences and their impact on grain yield of super hybrid rice under different irrigation conditions

PLOS ONE

Dear Dr. Zhou,

Thank you for submitting your manuscript to PLOS ONE. After careful consideration, we feel that it has merit but does not fully meet PLOS ONE’s publication criteria as it currently stands. Therefore, we invite you to submit a revised version of the manuscript that addresses the points raised during the review process.

Specifically, you need to correct the units of the grain yield results, which are often written as g m-1 instead of g m-2 (i have seen many instances in the text, in Table 2 also. Once this is completed the paper will be accepted.

We look forward to receiving your revised manuscript.

Kind regards,

Vincent Vadez

Academic Editor

PLOS ONE

Additional Editor Comments (if provided):

Please correct the unit of the grain yield results, which are often written as g m-1 instead of g m-2 (i have seen many instances in the text, in Table 2 also.

---

## [Author Response · Author response to Decision Letter 1]

21 Nov 2020

1. Please correct the unit of the grain yield results, which are often written as g m-1 instead of g m-2 (i have seen many instances in the text, in Table 2 also.

The error has been revised.

---

## [Editor Report · Decision Letter 2]

24 Nov 2020

Plant–atmosphere and soil–atmosphere temperature differences and their impact on grain yield of super hybrid rice under different irrigation conditions

PONE-D-20-26758R2

Dear Dr. Zhou,

We’re pleased to inform you that your manuscript has been judged scientifically suitable for publication and will be formally accepted for publication once it meets all outstanding technical requirements.

Kind regards,

Vincent Vadez

Academic Editor

PLOS ONE
---

## [Editor Report · Acceptance letter]

9 Dec 2020

PONE-D-20-26758R2 

Plant–atmosphere and soil–atmosphere temperature differences and their impact on grain yield of super hybrid rice under different irrigation conditions 

Dear Dr. Zhou:

I'm pleased to inform you that your manuscript has been deemed suitable for publication in PLOS ONE. Congratulations! Your manuscript is now with our production department. 

Kind regards, 

on behalf of

Dr. Vincent Vadez 

Academic Editor

PLOS ONE